# Arab2Vec: An Arabic word embedding model for use in Twitter NLP applications

**Abdelrahman Hamdy**[1], **Ayman Youssef**[2], **Conor Ryan**[3]*

**1** The Open University, Milton Keynes, United Kingdom, **2** Department of Computers and Systems, Electronics Research Institute, Cairo, Egypt, **3** Biocomputing and Developmental Systems (BDS) Research Group, University of Limerick, Limerick, Ireland

* conor.ryan@ul.ie

**Data availability statement:** All our models are available from this repository: https://github.com/Abdelrahmanrezk/AraETEWordVec.

## Abstract

The analysis of Arabic Twitter data sets is a highly active research topic, particularly since the outbreak of COVID-19 and subsequent attempts to understand public sentiment related to the pandemic. This activity is partially driven by the high number of Arabic Twitter users, around 164 million. Word embedding models are a vital tool for analysing Twitter data sets, as they are considered one of the essential methods of transforming words into numbers that can be processed using machine learning (ML) algorithms. In this work, we introduce a new model, *Arab2Vec*, that can be used in Twitter-based natural language processing (NLP) applications. Arab2Vec was constructed using a vast data set of approximately 186,000,000 tweets from 2008 to 2021 from all Arabic Twitter sources. This makes Arab2Vec the most up-to-date word embedding model researchers can use for Twitter-based applications. The model is compared with existing models from the literature. The reported results demonstrate superior performance regarding the number of recognised words and F1 score for classification tasks with known data sets and the ability to work with emojis. We also incorporate skip-grams with negative sampling, an approach that other Arabic models haven't previously used. Nine versions of Arab2Vec are produced; these models differ regarding available features, the number of words trained on, speed, etc. This paper provides Arab2Vec as an open-source project for users to employ in research. It describes the data collection methods, the data preprocessing and cleaning step, the effort to build these nine models, and experiments to validate them qualitatively and quantitatively.

## Introduction

Arabic Twitter analysis is an important research topic as Twitter contains many Arabic tweets. Around 164 million active monthly users are estimated to constantly add to the total [15]. However, working with the Arabic language is challenging for two reasons: first, there is a low number of data sets and pre-trained models, and second, the language itself is more complex to work with than Western languages. For example, the Arabic language contains diacritics that change the meaning of sentences. For example, سكر means sugar in Arabic. However, if we add diacritics سَكَر it means "got drunk".

**Funding:** The author(s) received no specific funding for this work.

*Word embedding models*, such as the popular and successful Word2Vec model[3], are responsible for converting words into vectors of numbers suitable for ML models. These vectors capture the semantic properties of words, which can help ML models do a better job in their required tasks. The models require training with huge text data sets to extract hidden relationships between words in the text. This enables them to produce word embedding models to help integrate text data into ML and deep learning applications. This has many applications in NLP, such as sentiment analysis [2], textual entailment [14], information retrieval [5], and question answering [4].

In this work, we propose a novel open-source Arabic word embedding model trained on a large dataset of approximately 186 million tweets. Our model compares favourably with existing models in the literature, achieving a vocabulary size of 2,027,042 words, significantly higher than the previous best of 1,476,715. A key strength of our model is its ability to recognise words that earlier models miss, particularly those related to COVID-19, as it is trained on more recent data. Additionally, it effectively handles emojis, treating them interchangeably with words, and recognises English words commonly found in Arabic tweets, such as "it" or "A+," which prior models typically ignore. We evaluate the model on two datasets: a COVID-related dataset[17] and the Arabic Sentiment Twitter Dataset (ASTD) [13]. In summary, our model addresses several challenges that earlier Arabic word embeddings faced, including limited vocabulary, poor emoji handling, and an inability to process embedded English or COVID-19-related terms, offering significant improvements across all these areas.

The proposed model has several advantages over other models as it can be seen in Table 1 in the literature. These can be summarised as follows:

- It is an open-source model trained on 186M tweets and, therefore, can recognise more words (2,027,042, approximately 33%) than previous models;
- It recognises COVID-19-related words and emojis with higher accuracy as it was trained on a newer data set;
- It has a variant trained using negative sampling. This is an advantage for our work, as we will see from the results that this variant achieves higher accuracy than the previous model.
- It exhibits better performance than the two state-of-the-art models from the literature (AraVec [1], AraWordVec [9] ) on the two tested data sets (COVID-19, ASTD).

The paper is organized as follows. We start with the Literature Survey section, which contains a literature survey on the research topic, before moving to the Background Information section, which describes the background knowledge needed to understand the proposed research. The paper then moves to the methodology section that describes the main steps of the proposed framework. The following section discusses the computational effort in the steps discussed above. Next, section provides a qualitative comparison between the model and two other well-known models from the literature using publicly available data. Next, Section presents a quantitative experimental comparison between the proposed model and models from the literature. Then the paper introduces the discussion section that summarizes the

**Table 1. Comparison Table between different embedding models.**

| Model | Number of recognized words | recognizing emojis | NS variant |
|---|---|---|---|
| Arab2Vec | 20,027,042 | Yes | Yes |
| AraVec | 1,476,715 | No | No |
| AraWordVec | 49,555 | No | No |

main findings of the work. Finally last section provides some conclusions and describes some future work.

## Literature survey

Here, we briefly introduce some related work related to our proposed work and some interesting applications from the literature. In [1], the authors proposed a pre-trained word embedding model in the Arabic context using the Word2Vec model. The trained models are general, distributed word embeddings trained on text-based data collected from the Internet, Wikipedia, and Twitter. They proposed six different models for the three different Arabic content domains. The Twitter model was trained using 77,600,000 Arabic tweets from 2016 to 2018. This model is used in our work to compare it with our results; we refer to it as the *AraVec* model. A set of word embedding models specifically for Twitter data was introduced by [9]. The authors compared their model with previous models from the literature and demonstrated the superior performance of their model in a word similarity task. We also compare them with this set of models and refer to them as the *ArWordVec* model. ArWordVec was also tested on a multi-class sentiment analysis task. The results show that AraVec achieved the best accuracy of 68.93, and ArWordVec achieved the best accuracy of 69.93. However, as our comparisons will show, the model can only recognise a relatively small number of words.

A multilingual word embedding model was introduced by [10]. This was trained with more than 100 languages, using their corresponding Wikipedia sites. The model showed superior performance compared to the then state-of-the-art English models. In [11], the authors trained an embedding model on 157 languages and evaluated it on ten languages. The results show good performance when compared with previous models. ArbEngVec [8] is a bilingual English-Arabic word embedding model. The model was trained on an enormous data set of 93 million sentence pairs extracted from the Open Parallel Corpus Project (OPUS), containing 90 languages and more than 2.7 billion parallel sentences.

The model was successfully evaluated using both *intrinsic* and *extrinsic* evaluations, where intrinsic evaluation tests the quality of the model in general, independent of a specific task, while extrinsic evaluation tests model quality when undertaking a specific NLP task. In [6], a word embedding model for medical and health applications in Arabic was introduced. Three trained models (Word2Vec, fastText, and Glove) were compared and evaluated, with the results showing a superior performance of Word2Vec and fastText [16] over the Glove model [7].

These pre-trained models were used in different NLP applications. In [19], a convolution neural network (CNN) was trained using the Twitter continuous bag of words model (AraVec Model). This CNN was used for an Arabic tweet sentiment classification task. In [20], AraVec models generate the feature vectors for words (tokens from tweets). These feature vectors are used for training a long short-term memory (LSTM) model for emotion analysis of Arabic tweets. When their technique was compared to a Support Vector Machine (SVM), a Random Forest (RF), and a fully connected deep NN, it produced the best performance results and increased validation by 9% over the previous best SVM result. In [21], the authors used AraVec and AraWordVec for text classification for Arabic social media tweets. In this application, AraWordVec achieved higher performance than AraVec. The authors employed three machine learning classification algorithms: RF, SVM, and Gaussian Naive Bayes. In [25], 26 different text pre-processing techniques were used to treat Arabic tweets before creating a classifier to identify tweets containing health-related information. The author's analysis aimed

to determine how pre-processing methods influenced traditional algorithms. They discovered that most strategies did not improve the accuracy of the base model. They used different AraVec and AraWordVec models and tested them on two data sets. In [24], the authors used the pre-trained AraWordVec to build their methodology for sentiment analysis of the Arabic language.

## Background information

Word2vec is a word representation method with numerous uses in many NLP applications. Initially proposed by Mikolov and his team at Google, Word2Vec converts each input token (word) to a real-valued vector. Similar words that appear in the same context are assigned similar real-valued vectors (similar representation). Word2vec is a deep neural network model that takes the tokens as inputs and produces the vector representations. Word representation similarities yield semantic features, frequently sustaining semantic linkages in vector operations on word vectors. For instance, the vector of the (King) plus the (Woman) minus the (Man) is not far from the vector of (Queen) [18].

In this paper, we use three different variants of Word2Vec models. In the following subsections, we discuss briefly the differences between these variants. Fig 1 shows the block diagram of Skip-gram and CBOW algorithms.

### Continuous Bag of Words (CBOW)

In this model, we train a neural network to predict a target word from its neighbouring words, essentially predicting the word from context. The neighbouring words could be a single word or several words. The order of context words makes no difference, hence the name *bag of words*. A limitation of CBOW is that it gives all context words equal importance when making

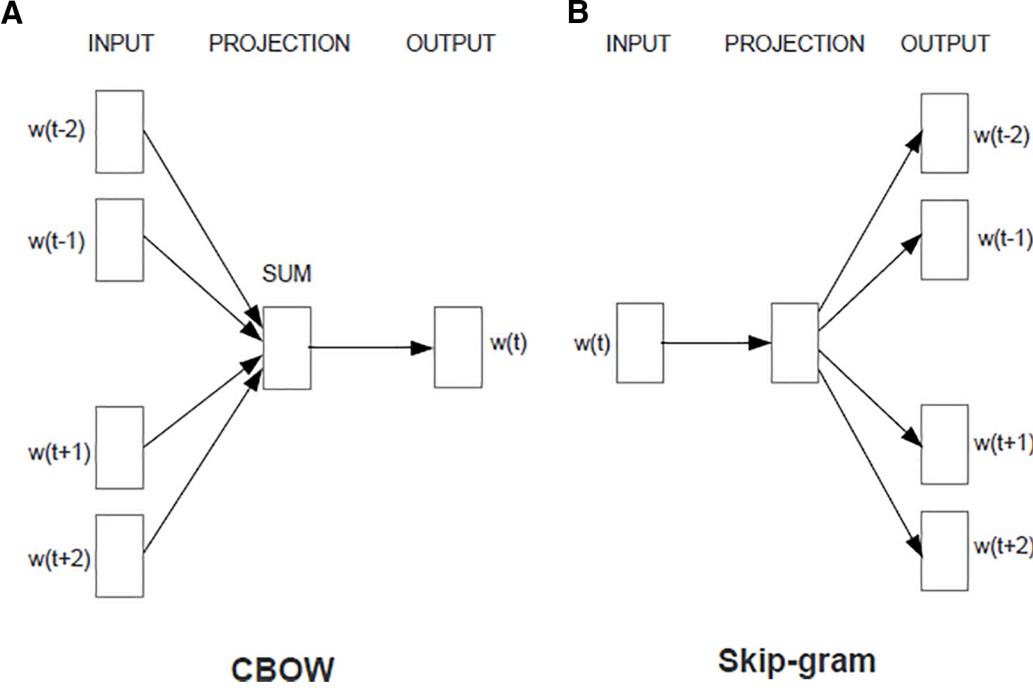

**Fig 1. Word2Vec models** [3].

a prediction [22]. However, this is not the case in practice since some words have a higher predictive value.

### Skip-Gram (SG)

An alternative model is the Skip-gram, essentially the opposite of the CBOW model. This model tries to detect the context of the sentence based on the input word.

CBOW is generally faster to train than Skip-gram, but Skip-gram can work better with small data sets [3].

### Skip-Gram With Negative Sampling (SGNS)

Skip-gram models can take a long time to train due to the large number of required neuron weights. To mitigate this, the Skip-gram creators proposed a technique called "negative sampling" to address this problem. In a typical neural network, training involves modifying the neural weights to improve the accuracy of each training sample. However, when using negative sampling, only a tiny portion of the weights are modified with each sample.

## Proposed methodology

Fig 2 shows the proposed framework of this paper.

## Data collection

Twitter data has become increasingly important and attractive to NLP researchers in recent years.

This is partly because tweets can easily be queried using a developer account on Twitter and also because they have tags for location and other user information. Our data set comprises 377,000,000 tweets collected between 2008 and 2021 from all Arabic Twitter sources.

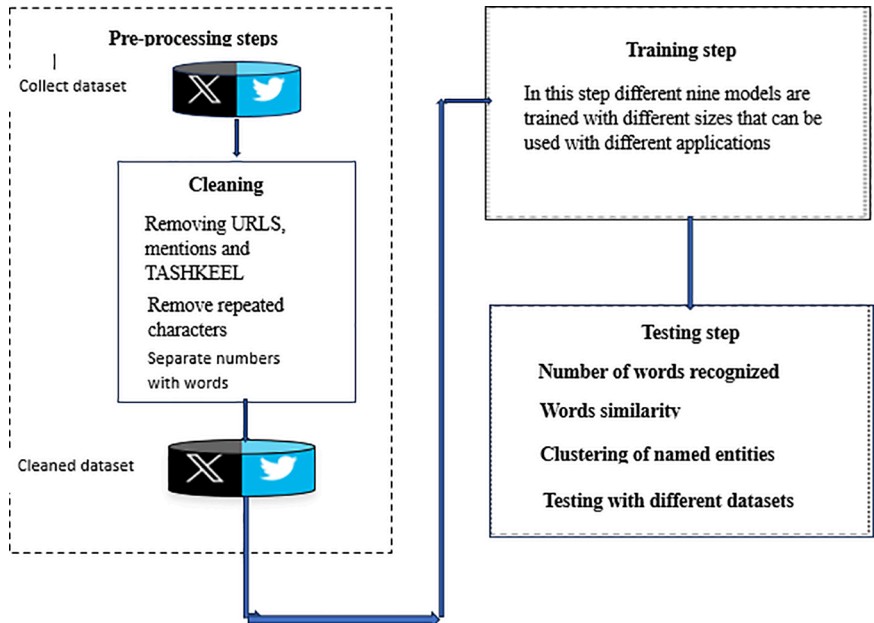

**Fig 2. The proposed framework.**

We gather tweets by examining the language information provided for each one, specifically from a section called the "language column." This information is used to select tweets. The data is used for research purposes only, and the data analysis complies with Twitter's terms and conditions. We process this data set to produce useful data to train the models.

## Data pre-processing

Pre-processing the data is a crucial step in any NLP model to prepare the data for the word embedding model to process. The pre-processing steps we undertook included the following:

### Remove duplicate Tweets and filter Tweets

We first filtered the tweets to remove duplicates, as they increased the model training time without benefit. This reduced the data set size from 377,000,000 to 186,000,000 tweets.

### Cleaning Arabic text

There were also several tasks specific to Arabic text that were required to clean the text:

- Replace URLs with a single Arabic word.
  Some tweets may contain a URL for a site. These URLs are removed and replaced with رابطويب, which is Arabic for *web site link*.
- Replace *mentions* with a single Arabic word.
  A *mentions* in Twitter is a reference to a person or a brand; we remove these as there is nothing to learn for the embedding model from these words and replace them with the word حسابشخصي, which is an Arabic word meaning *personal account*.
- Remove diacritics (Tashkeel).
  Arabic uses diacritics to change the pronunciation of words. We found that most tweets don't contain diacritics, so we removed them before training the model.
- Remove characters repeated more than two times sequentially.
  Some tweets contain repeated characters, such as هيييييه and سلااااام , meaning hhhhhhh (akin to *lol* or some indication of amusement/laughter) and peace, respectively. These cases are edited by removing repeated characters so that if a character is repeated more than two times in a word, all but the first occurrence is removed.
- Separate numbers associated with words.
  Some tweets may contain a word that has a number attached to it. In this step, we remove any numbers connected to a word.
- Reduce repeated emojis used sequentially.
  Some Twitter users use the same emoji multiple times in sequence in a tweet to indicate a particularly strong emotion. This step removes this sort of repetition.
- Remove extraneous spaces.
  Some tweets may contain more than one space sequentially. In this step, we remove this repetition.

After these steps, tokenisation using the *treebank* library from Python is applied.

## Computational effort in these steps

This section discusses the time consumed in each of the previous steps.

The training of the models took around four days per model, which varied from 75 to 85 hours for each; we produced nine models in total. These comprise three tri-gram models

(CBOW, Skip-gram, and Skip-gram with negative sampling) with a word count threshold (the number of words examined when calculating a context) of 300; another three tri-gram models (CBOW, Skip-gram, Skip-gram with negative sampling) with a word count threshold of 100; and the third three uni-gram models (CBOW, Skip-gram, Skip-gram with negative sampling) with a word count threshold of 100. Tri-gram models process text three words simultaneously, while uni-grams do so one word at a time. In this work, we use tri-grams because they allow the model more freedom in training by not limiting it to smaller n-grams. We do this to make different models with different sizes and features available for the Arabic NLP research community. (The source code and all models are downloadable from the following link: https://github.com/Abdelrahmanrezk/AraETEWordVec)

## Model evaluation and comparison

We compare the performance of our model and that of the AraVec and ArWordVec models in four ways:

- The total number of words recognised, which depends on the number of tweets that the model is trained on;
- Word similarity, which shows what words the model finds similar (and with what similarity score) to an input word;
- Clustering of positive and negative words (in this step, we test the model with a subset of words that can be clustered into two groups, negative and positive, and we draw the output representation from the model to see if similar words are getting representations that are close to each other or not);
- Clustering of named entities (in this step, we choose a random set of named entities that fall under different categories, such as person and country names, and see if the model can cluster them into similar groups).

### Number of words recognized

We initially compare our model with others based on the number of recognised words. As seen in Table 2, our model can recognise many more than the others.

### Word similarity

In this section, we examine the performance of each model when trying to suggest similar words to a random selection of words. We start with the word خريج, which means *graduate* in English. Fig 3 shows the results obtained from each of the three models. In Fig 3, the bar on the left shows how similar the model believed a particular suggested word is, while the word itself is next (in Arabic), followed by its English translation.

The results show that all three models successfully recognise the word. It can be seen that both ArWordVec and AraVec produce five words that are syntactically related to the word, like خريجين, which means graduate. Arab2Vec, on the other hand, produced more words that

**Table 2. Number of recognised words.**

| Model | Number of recognized words |
|---|---|
| Arab2Vec model | 2,027,042 |
| ArVec model | 1,476,715 |
| ArWordVec model | 49,555 |

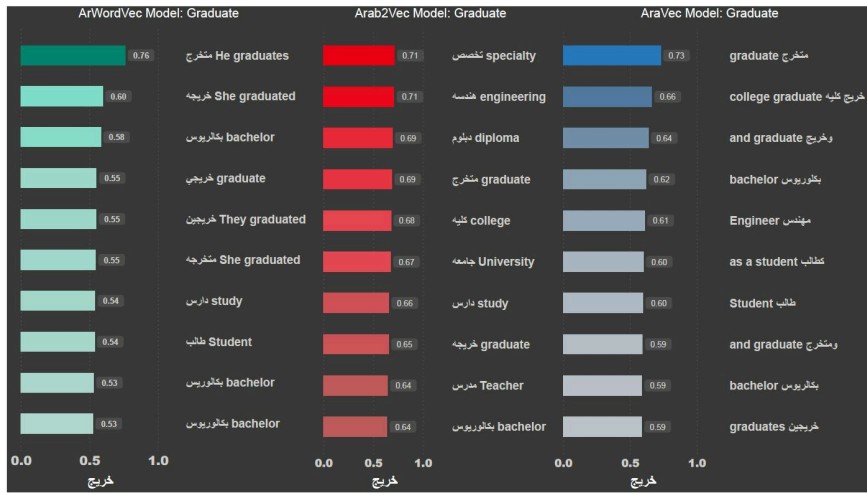

**Fig 3. Words similar to "Graduate" suggested by the three models.**

are semantically related to the word, like تخصص, which means speciality, كليه, which means college and جامعه, which means university. The second word we use to test the three models is بنغازي (Benghazi), which is a city in Libya. Fig 4 shows the most similar words to the word Benghazi. The word was not recognised using the AraWordVec model, while the ArVec and Arab2Vec models had similar results. Next, we test the model using the pink flower emoji; Fig 5 shows the most similar emojis to the pink flower emoji 🌸 .

The ArWordVec model was not able to recognise the *pink flower* emoji, while the AraVec model produces words that are not relevant to it. The Arab2Vec model, on the other hand,

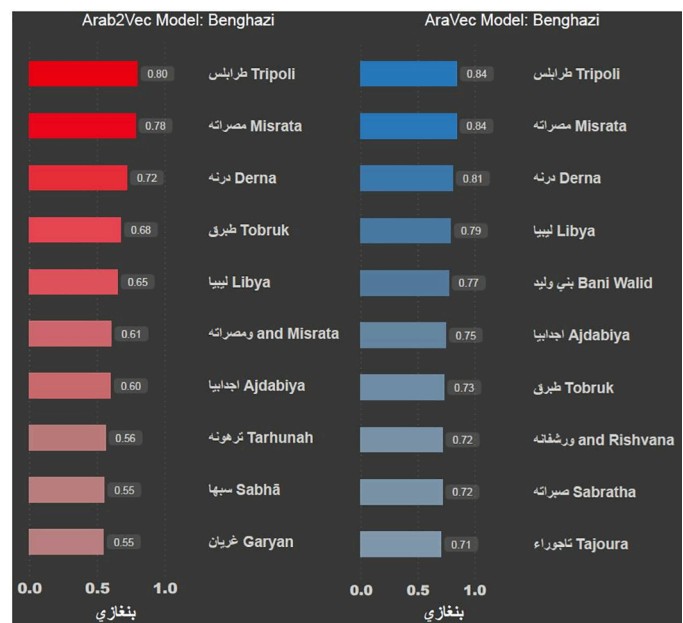

**Fig 4. Words similar to "Benghazi" suggested by the AraVec Model and Arab2Vec Model.**

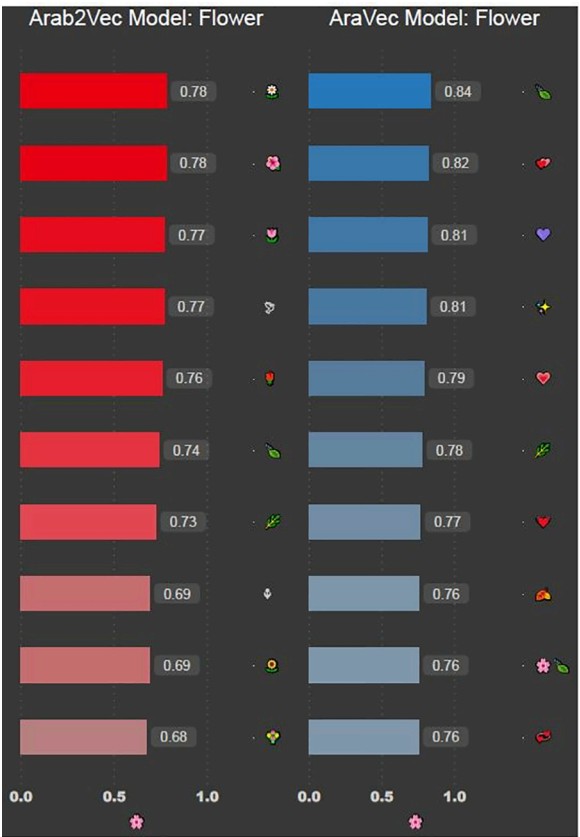

**Fig 5. Words similar to 🌸 emoji for AraVec and Arab2Vec model.**

successfully produced several emojis that are similar in meaning to the *pink flower* emoji 🌸 . Next, we test the model with the *rolling-on-the-floor laughing* emoji🤣 as shown in Fig 6.

The Arab2Vec model is the only model that can recognise Emojis with good accuracy, and the other two models(AraVec and ArWordVec) failed to recognise that emoji.

## Clustering of positive and negative words

This exercise will see if our model can capture the similarities between words from different clusters. In this test, introduced by [1], we test the model with a small subset of words from two clusters (negative and positive) to see if the model can capture the similarities between the words. The words we use are provided in Table 3:

Fig 7 shows the clustering of positive and negative word results for the AraVec model. Fig 8 shows the clustering of positive and negative word results for the Arab2Vec model. It can be seen in Fig 8 that the Arab2Vec model could cluster the words correctly into two classes.

ArWordVec could not cluster these words because the model is trained on fewer words.

## Clustering of named entities

In this section, we compare the models using different named entities. The entities we use are names from countries, people, months, social media platforms, different organisation types,

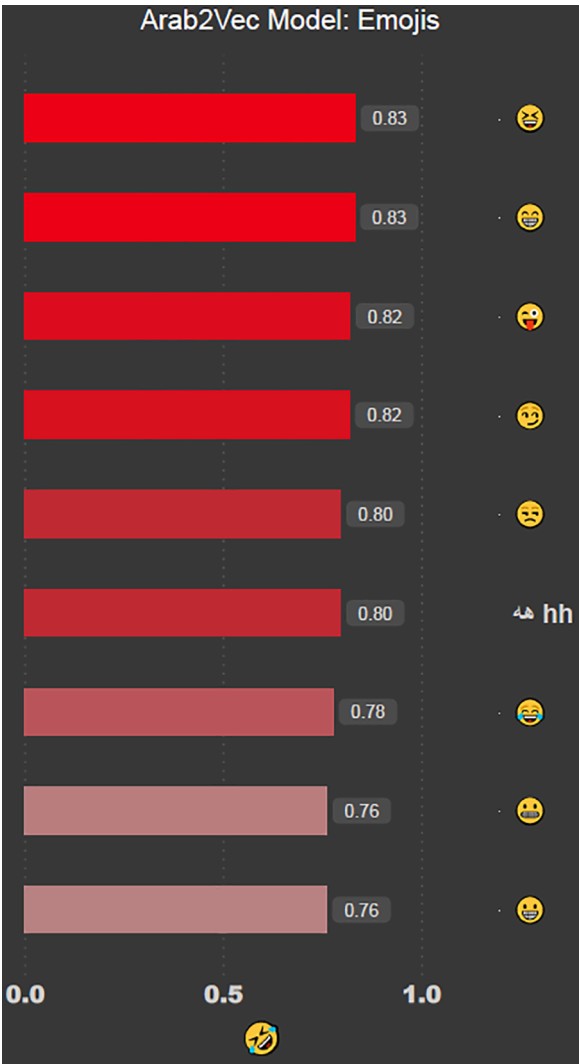

**Fig 6. Words that are similar to the 🤣 emoji as produced by the Arab2Vec model**. Recall that "hh" is often used in Arabic to denote amusement or laughter.

**Table 3. Set of positive and negative words.**

| | | | |
|---|---|---|---|
| موهوب | Gifted | قواد | pimp |
| مذهل | amazing | سافل | varmint |
| ممتاز | excellent | جاسوس | spy |
| رائع | wonderful | عميل | agent |
| متكامل | complete | خائن | traitor |
| متطور | developed | مطبل | hypocrite |
| خرافي | fabulous | قذر | dirty |
| رهيب | awesome | مرتزق | mercenary |
| مهاري | skilled | خسيس | miserly |
| - | | مخنث | sissy |

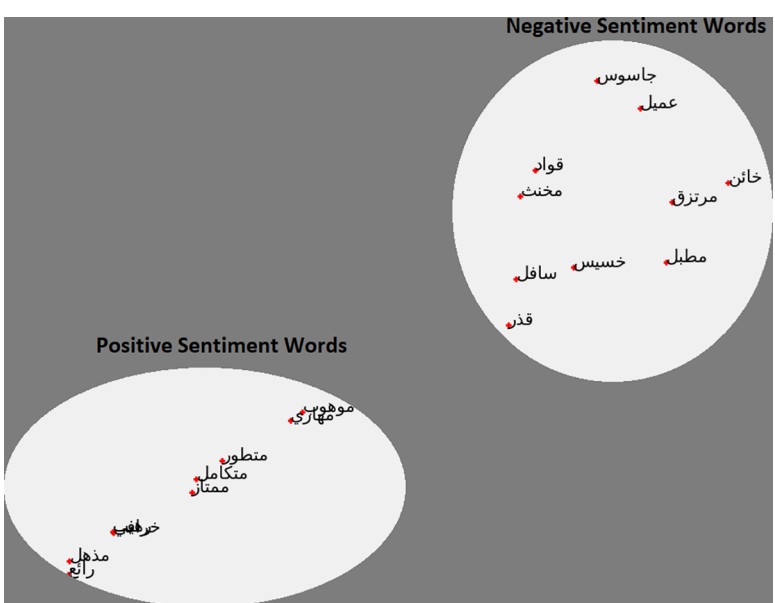

**Fig 7. Words clustering from the AraVec model.** The positioning of the words indicates two clear clusters.

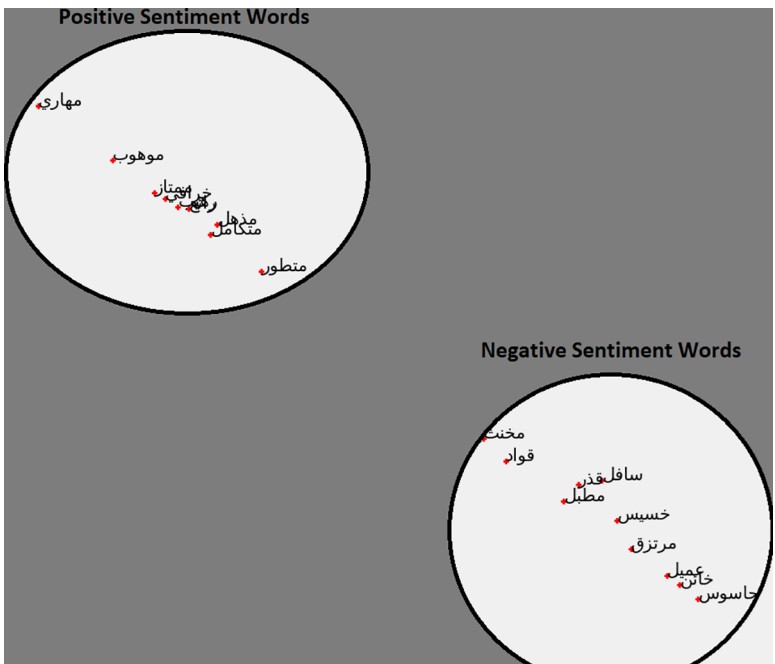

**Fig 8. Word clustering Arab2Vec model.** The arrangement of the words indicates that no meaningful clustering has taken place.

internet-related devices, electronic company names, military-related equipment and vehicle types. Several words are chosen randomly to represent each entity. The same words are used to compare our model with previous models to ensure fairness in comparison. The words we

use are provided in the table in the same order mentioned in the text. Table 3 shows the words used in this experiment and their clustering entity names.

Fig 9 shows the clustering of named entities results from the Arab2Vec model. In contrast, Fig 10 shows the clustering of named entities results from the AraVec model.

It can be seen from the figures that Arab2Vec was able to cluster the named entities into separate regions, while we see that in AraVec results, the two clusters of electronic companies and internet devices interfere with each other and become inseparable by this model. Table 4 shows the different name entities used in this experiment along with there Arabic translation.

The results couldn't be compared with the ArWordVec model because it couldn't recognise some words as it was trained on a few words.

## Experimental comparison of the models

In this section, we test our proposed models experimentally to calculate the accuracy difference between the models proposed in the literature and our model. These comparisons are conducted using publicly available datasets. Note that it is not possible for us to redistribute the training data as this would violate copyright.

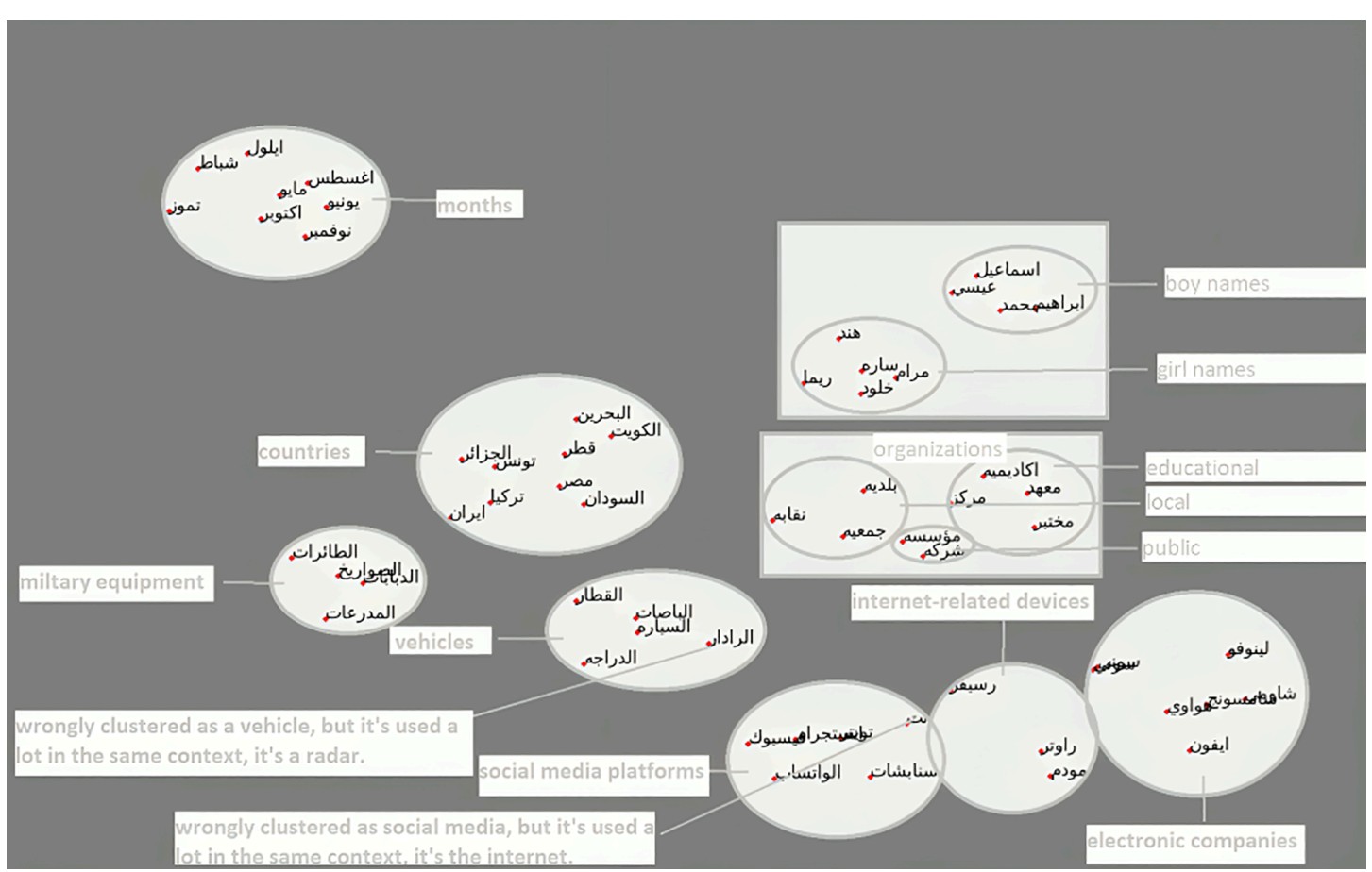

**Fig 9. Clustering of named entities by the Arab2Vec model.**

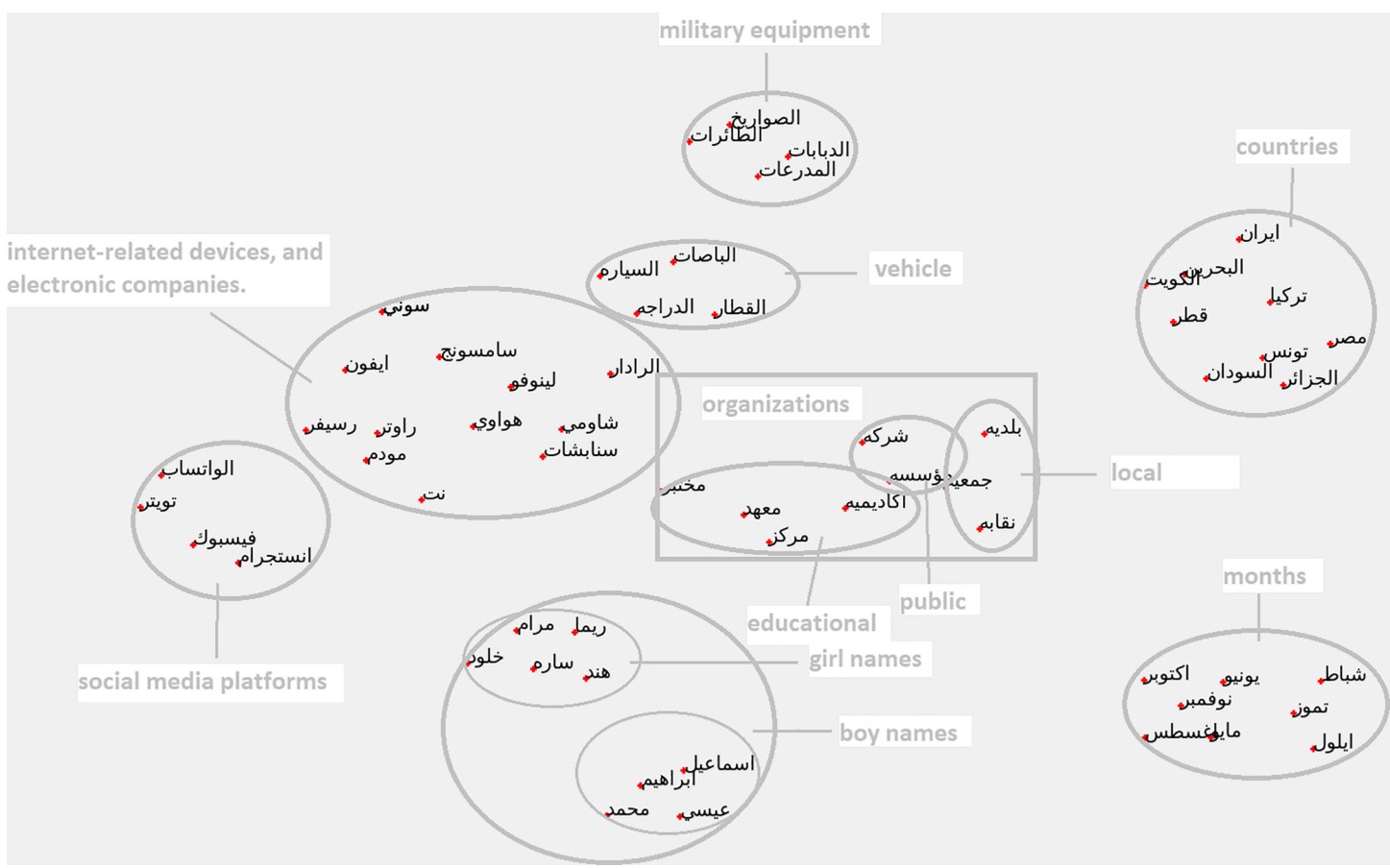

**Fig 10. Clustering of named entities AraVec model.**

## COVID-19 data set classification

In this problem, we use 10,000 tweets classified as COVID-19-related and non-COVID-19-related tweets; this data set [26] is freely available for download from (https://github.com/SarahAlqurashi/COVID-19-Arabic-Tweets-Dataset).

In this experiment, we use a set of ML models (logistic regression, support vector classification (SVC), AdaBoost and Gradient Boost (GB)) and compare their F1 scores using the maximum F1 score achieved by each model. These experiments were conducted with each of Arab2Vec, AraVec and ArWordVec. The AdaBoost algorithm usually gives us the highest F1 score, as shown in Table 5. We also employ a deep learning model (LSTM with batch normalisation, the hyper-parameters shown in Table 5). Table 6 shows a comparison between the different models. The second column shows the F1 scores of an ML model (AdaBoost), while the third column shows the F1 scores of the deep learning models. Table 7 shows the a comparison between F1 scores of the best performance ML or DL models on COVID dataset.

The results show that our model achieves higher accuracy in classifying COVID-19 data sets with both ML (AdaBoost) and deep learning models.

**Table 4. Named entities table.**

| Entity type | Arabic | English translation | Arabic | English translation |
|---|---|---|---|---|
| countries | تونس | Tunisia | الجزائر | Algeria |
| | السودان | Sudan | قطر | Qatar |
| | الكويت | Kuwait | البحرين | Bahrain |
| | تركيا | Turkey | مصر | Egypt |
| | ايران | Iran | | |
| people | محمد | Muhammad | ابراهيم | Abraham |
| | اسماعيل | Ismail | هند | Hend |
| | ساره | Sarah | مارام | Maram |
| | عيسي | Aisa | خلود | Kholod |
| | ريما | Rima | | |
| months | أغسطس | August | يوليو | June |
| | ايلول | September | اكتوبر | October |
| | شباط | February | نوفمبر | November |
| | مايو | May | تموز | July |
| social media platforms | فيسبوك | Facebook | انستجرام | Instagram |
| | تويتر | Twitter | سنابشات | Snapchat |
| | واتساب | WhatsApp | | |
| organization types | جمعيه | Charity association | معهد | Institute |
| | اكاديميه | academy | شركه | company |
| | مختبر | lab | مركز | center |
| | بلديه | syndicate | نقابه | municipality |
| internet-related devices | مودم | router | راوتر | Modem |
| | نت | Net | ريسيفر | receiver |
| electronics companies | ايفون | iPhone | هواوي | Hawaii |
| | سامسونج | Samsung | لينوفو | Lenovo |
| | سوني | Sony | شاومنج | Xiaomi |
| military equipment | الدبابات | tanks | الطائرات | plans |
| | المدرعات | armor | الصواريخ | rockets |
| types of vehicles | الدراجه | cycle | السياره | car |
| | القطار | train | الباصات | buses |
| | الرادار | radar | | |

**Table 5. Deep learning model parameters of LSTM.**

| | |
|---|---|
| hidden later number of neurons | 25 |
| learning rate | .00005 |
| epochs | 10 |
| optimizer | RMsprop |

**Table 6. F1 score of different ML algorithms on COVID-19 test dataset.**

| Word2Vec model | LR F1 | SVC F1 | AdaBoost F1 | GB F1 |
|---|---|---|---|---|
| AraVec CBOW | .84 | .79 | .84 | .72 |
| Arab2Vec CBOW | .84 | .82 | .9 | .72 |
| ArWordVec CBOW | .779 | .75 | .78 | .7 |
| AraVec Skip-gram | .9 | .87 | .89 | .7 |
| Arab2Vec Skip-gram | .89 | *.91* | .92 | .73 |
| ArWordVec Skip-gram | .85 | .84 | .84 | .73 |
| *Arab2Vec Skip-gram NS* | .89 | .89 | .93 | .73 |

## ASTD classification

The Arabic sentiment tweets data set (ASTD) is a data set of four classes [13] (objective, subjective positive, subjective negative and subjective neutral). This data set is also freely

**Table 7. F1 score of different models on the COVID-19 test data set using machine learning (ML) and deep learning (DL). The overall best performer, Arab2vec Skip-gram NS, is in italics.**

| Word2Vec model | ML F1 | DL F1 |
|---|---|---|
| ArWordVec CBOW | .86 | .9 |
| AraVec CBOW | .88 | .87 |
| Arab2vec CBOW | .9 | .93 |
| ArWordVec Skip-gram | .83 | .87 |
| AraVec Skip-gram | .85 | .89 |
| Arab2Vec Skip-gram | .87 | .91 |
| *Arab2vec Skip-gram NS* | *.92* | *.93* |

downloadable (https://github.com/mahmoudnabil/ASTD). As with the previous experiment, we first used the same set of machine learning models (logistic regression (LR), support vector classification (SVC), AdaBoost and Gradient Boost (GB)) and the F1 score was compared. It can be seen in Table 8 that the AdaBoost model gives the best F1 score, so we use it in our comparisons. Table 9 compares each model's machine learning F1 scores and deep learning F1 scores. In the first column, we compare AdaBoost's F1 score using different models, while in the second column, we compare deep learning models (LSTM with batch normalisation) DL F1. The distribution of these tweets is shown in Table 8. Table 10 shows the comparison between the F1 scores of the best performance ML or DL models on ASTD dataset.

The results show that our model performs better than others in training DL models. However, the ArWordVec CBOW model achieves higher accuracy in the ML models.

**Table 8. Distribution of Arabic tweets in the ASTD dataset.**

| Objective | 6691 |
|---|---|
| Subjective Positive | 799 |
| Subjective Negative | 1684 |
| Subjective Neutral | 832 |

**Table 9. F1 score of different ML algorithms on the ASTD test dataset.**

| Word2Vec model | LR F1 | SVC F1 | AdaBoost F1 | GB F1 |
|---|---|---|---|---|
| AraVec CBOW | .56 | .55 | .62 | .51 |
| Arab2Vec CBOW | .62 | .59 | *.71* | .53 |
| ArWordVec CBOW | .58 | .57 | .62 | .55 |
| AraVec Skip-gram | .61 | .57 | .67 | .54 |
| Arab2Vec Skip-gram | .59 | .57 | .63 | .53 |
| ArWordVec Skip-gram | .61 | .56 | .65 | *.57* |
| Arab2Vec Skip-gram NS | *.62* | *.62* | .66 | .5 |

**Table 10. F1 score of different models on ASTD test dataset.**

| Word2Vec model | ML F1 | DL F1 |
|---|---|---|
| AraVec CBOW | .65 | .63 |
| Arab2Vec CBOW | .67 | .66 |
| ArWordVec CBOW | .69 | .61 |
| ArWordVec Skip-gram | .62 | .65 |
| AraVec Skip-gram | .64 | .67 |
| Arab2Vec Skip-gram | .65 | .67 |
| Arab2Vec Skip-gram NS | .67 | *.7* |

## Discussion

*Word embedding models* are key components of most NLP applications. Building Arabic embedding models presents unique challenges due to the complexities of the Arabic language. In this work, a novel Arabic word embedding model is proposed. The proposed word embedding models are trained on 186 million tweets, resulting in a model that recognises 33% more words than previously available models in the literature. Furthermore, the proposed model can recognise COVID-related words and emojis, providing an additional advantage over prior models. This work offers NLP researchers and practitioners nine distinct models of varying sizes tailored for different Arabic NLP applications. One of these models is trained using negative sampling, representing a novel contribution that has not been previously available in the literature. The proposed models are evaluated in four ways: the total number of recognised words, word similarity, clustering of positive and negative words, and clustering of named entities. Experimental results indicate that the proposed models demonstrate significant performance improvements over existing models such as AraVec and ArWordVec.

An alternate approach would be to employ a Large Language Model (LLM) to perform this task. In particular, LLMs' ability to create context-aware representations is attractive. However, a dedicated word embedding model like the one proposed here can reasonably be expected to be more efficient and resource-friendly than fine-tuning a large language model for word embedding tasks. Future work will explore this trade-off further.

## Conclusion

A novel word embedding model for Arabic, Arab2Vec, is introduced. The model is trained on a vast dataset and demonstrates superior performance compared to the state-of-the-art models, AraVec and ArWordVec.

The Arab2Vec model was evaluated and compared against AraVec and ArWordVec in four aspects: the total number of recognised words, word similarity, clustering of positive and negative words, and clustering of named entities. Additionally, the model was tested experimentally on two distinct datasets: COVID-19 and ASTD.

Arab2Vec offers several advantages over existing models. First, it can recognise almost two million words, compared to the previous high of 1,476,715. Second, it effectively handles new terms introduced on Twitter, such as COVID-19-related vocabulary. Third, unlike other models in the literature, it can process emojis, broadening its utility in social media analysis.

Another key contribution of this work is including models trained using negative sampling, which has not been previously introduced in AraVec or ArWordVec. Arab2Vec is being made available to the NLP and AI communities for research purposes, with all source code and models offered freely.

We plan to train additional word embedding models, such as GloVe and fastText, and compare our approach with alternative methods, including transformer-based ones.

## Author contributions

**Conceptualization:** Ayman Youssef, Conor Ryan.

**Data curation:** Abdelrahman Hamdy.

**Investigation:** Abdelrahman Hamdy, Ayman Youssef, Conor Ryan.

**Methodology:** Abdelrahman Hamdy, Ayman Youssef, Conor Ryan.

**Software:** Abdelrahman Hamdy.

**Supervision:** Ayman Youssef, Conor Ryan.

**Validation:** Abdelrahman Hamdy.

**Visualization:** Abdelrahman Hamdy.

**Writing – original draft:** Ayman Youssef, Conor Ryan.

**Writing – review & editing:** Ayman Youssef, Conor Ryan.

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
