## [Decision Letter · Decision Letter 0]

14 Mar 2023

PONE-D-23-01033Arab2Vec: An Arabic word embedding model for use in Twitter NLP applicationsPLOS ONE

Dear Dr. Ryan,

Thank you for submitting your manuscript to PLOS ONE. After careful consideration, we feel that it has merit but does not fully meet PLOS ONE’s publication criteria as it currently stands. Therefore, we invite you to submit a revised version of the manuscript that addresses the points raised during the review process.

ACADEMIC EDITOR: The paper lacks clarity in terms of its novelty. The main contribution of the paper needs to be explained clearly. Additionally, a workflow diagram for the proposed method needs to be included, outlining all the steps involved. The results of the proposed method should also be compared with state-of-the-art methods, and statistical analysis should be conducted. To improve the quality of the paper, it is important to explain all the results in detail with other previous methods and add a discussion section. Once these revisions are made, revised and resubmit the paper. 

We look forward to receiving your revised manuscript.

Kind regards,

Junaid Rashid, Ph.D

Academic Editor

PLOS ONE

3. In your Methods section, please include additional information about your dataset and ensure that you have included a statement specifying whether the collection and analysis method complied with the terms and conditions for the source of the data.

Reviewers' comments:

Reviewer's Responses to Questions

**Comments to the Author**

1. Is the manuscript technically sound, and do the data support the conclusions?

Reviewer #1: Yes

Reviewer #2: Yes

2. Has the statistical analysis been performed appropriately and rigorously? 

Reviewer #1: I Don't Know

Reviewer #2: N/A

3. Have the authors made all data underlying the findings in their manuscript fully available?

Reviewer #1: No

Reviewer #2: No

4. Is the manuscript presented in an intelligible fashion and written in standard English?

Reviewer #1: Yes

Reviewer #2: No

5. Review Comments to the Author

Reviewer #1: Thank you for this work. There is slight progress compared to the literature. The conclusions are supported by the results. The authors should be encouraged to submit a revised version addressing the following points:

1. (Abstract) Can you incorporate why Arabic Twitter data sets are particularly interesting to be highly researched?

2. The data collection and data pre-processing steps are well described, but how the Arab2Vec models were trained is very superficial. Which optimization algorithm and configurations did you use? How did you split training / validation sets for Tables 6 and 8? It would be helpful to include in the github repository the codes for scrapping and training of the models as well, not only the trained models. Please make more clear the difference between validation and test sets. For a non-experienced reader this might be unclear.

3. You mention that you trained nine Arab2Vec models, but Tables 5, 6, 8 and 9 only show three of

them. Please specify exactly which are being shown. Is there a reason why you don’t provide the

results for the other six models? I suppose this wouldn’t require too much extra effort.

4. Please include an English translated version of Table 2, Figs 6, 7, 8 and 9.

5. How did you choose 25 for the hidden dimension of the LSTM? Have you tried other values?

6. Section “Effort in these steps” could be renamed to “Computational effort in these steps”. You

also need to give the specifications of the system (GPU, processor, RAM, etc…).

7. Line 288 which is in bold. This information is very important and should be made clear much earlier in the paper.

8. Please revise the English:

a. Line 184: “of 100. where tri-gram”

b. Line 250: “vehicle types The”

c. Figs 8 and 9 captions: “named”

d. Table 5: “GB F!”

e. Line 280: “One again”

f. Line 302: "models we compare with Arab2Vec is being released"

Reviewer #2: I believe that the paper, as I understood it, meets the criteria to be published and thus, I would suggest revising it, albeit several points would need to be addressed. The paper is technically sound and meets the scientific standard. A number of details and questions are open but I believe they can be resolved and thus, I would proceed with the publication procedure.

The grammatical deficits stand out negatively and diminish my confidence in the author’s work as I cannot fully rely on the content which may be ambiguous due to grammatical errors. I strongly recommend the authors to seek help either from peers or professionals to improve the language of the paper such that reviewer and readers - in the case of acceptance - can focus on the content of your work.

You can find my complete review in the attachment with clear instructions what to address.

6. PLOS authors have the option to publish the peer review history of their article (what does this mean?). If published, this will include your full peer review and any attached files.

Reviewer #1: No

Reviewer #2: **Yes: **Auss Abbood

---

## [Author Response · Author response to Decision Letter 1]

22 Jun 2023

Reviewer 1

1. (Abstract) Can you incorporate why Arabic Twitter data sets are particularly interesting to be highly researched?

The reason that Arabic Twitter data sets are important is the large number of Arabic users on Twitter, and it is estimated that around 186 million Arabic users add to Twitter Arabic tweets.

This reason is mentioned in the introduction now it is added to the abstract.

2. The data collection and data pre-processing steps are well described, but how the Arab2Vec models were trained is very superficial. Which optimization algorithm and configurations did you use? How did you split training/validation sets for Tables 6 and 8? It would be helpful to include in the GitHub repository the codes for scrapping and training of the models as well, not only the trained models. Please make clearer the difference between validation and test sets. For a non-experienced reader this might be unclear.

The GitHub repository contains all the codes for scraping and training the models. We didn’t hide any piece of code. However, the repository was private in the first review round. In this round, we changed its state to public.

3. You mention that you trained nine Arab2Vec models, but Tables 5, 6, 8 and 9 only show three of

them. Please specify exactly which are being shown. Is there a reason why you don’t provide the

results for the other six models? I suppose this wouldn’t require too much extra effort.

There is no extra effort in running the other six models. However, to our understanding, the results of these three models are enough to prove the effectiveness of the proposed model.

4. Please include an English translated version of Table 2, Figs 6, 7, 8 and 9.

We added an English translation for Table 2; however, in our opinion, adding an Arabic translation to Figures 6,7,8 and 9 will make them messy. So, we made the following modifications:

1- Change the background colour to grey.

2- Added circles over each cluster to differentiate the words coming from each cluster to help non-Arabic speakers understand the figures.

We pointed out words that overlap between different clusters

5. How did you choose 25 for the hidden dimension of the LSTM? Have you tried other values?

These values were chosen depending on similar work from literature.

6. Section “Effort in these steps” could be renamed to “Computational effort in these steps”. You

also need to give the specifications of the system (GPU, processor, RAM, etc…).

Fixed the title of the section was changed. System information

7. Line 288 which is in bold. This information is very important and should be made clear much earlier in the paper.

8. Please revise the English:

a. Line 184: “of 100. where tri-gram”

Fixed where removed

b. Line 250: “vehicle types The”

Fixed

c. Figs 8 and 9 captions: “named”

Fixed

d. Table 5: “GB F!”

This is the abbreviation for gradient boost F1 score. Fixed

e. Line 280: “One again”

Fixed and changed to once again

f. Line 302: "models we compare with Arab2Vec is being released"

Fixed

Another contribution of this work is that we provide 288 models that were trained with negative sampling that wasn’t introduced in the previous 289 models from the literature (AraVec and AraWordVec). Arab2Vec is being released to the 290 NLP and AI communities for research purposes.

Reviewer 2

Review, 20 Feb. 2023

Manuscript Number: PONE-D-23-01033

Manuscript Title: Arab2Vec: An Arabic word embedding model for use in Twitter NLP applications

Summary

In this work, the authors present their work on learning distributed vector representation (also called word embeddings) on Tweets written in Arabic using the Word2Vec algorithm by Mikolov et al. Their work made several improvements compared to existing, similar word embeddings. Their word embeddings were trained on larger quantities and more recent Tweets compared to previous work, they learned embeddings on emojis, which none of the previous embeddings for Arabic tweets did, and used a variation of Word2Vec that improved training time and accuracy in the evaluation step.

Their embeddings were evaluated in comparison to existing embeddings in a qualitative and quantitative fashion. The evaluation consisted of: (i) Comparing the number of overall recognized words as a measure of completeness, (ii), qualitative assessments of word similarity in the embedding space coming from manually selected words, (iii) a visual assessment of clusters of word embeddings for negative and positive sentiment words, (iv) a visual assessment of the cluster quality of words belonging to the same named entity, (v) and finally testing the performance of several classification algorithms using several word embeddings to identify which embedding performs the best showing that their proposed embeddings yielded the best results in most tasks.

Overall feedback

+ The manuscript tells a clear and consistent story that is easy to follow

+ The authors presented a diverse and compelling evaluation strategy

+ They motivated and described their work in a succinct manner

+ I believe it is important that the authors worked on Arabic word embeddings since Arabic does not enjoy as much attention as other languages in the research community despite being spoken by relatively many people

- The paper has strong grammatical deficits

The grammar of the paper has been thoroughly revised by a native English speaker

- The section on Literature Survey is wordy and unstructured

- The models and code are not accessible with the provided GitHub link

There was a mistake that the Github link is still private. We currently changed it to public to allow reviewers to see the codes and models of our work.

I believe that the paper, as I understood it, meets the criteria to be published and thus, I would suggest revising it, albeit several points would need to be addressed. The paper is technically sound and meets the scientific standard.

The grammatical deficits stand out negatively and diminish my confidence in the author’s work as I cannot fully rely on the content which may be ambiguous due to grammatical errors. I strongly recommend the authors to seek help either from peers or professionals to improve the language of the paper such that reviewer and readers in the case of acceptance can focus on the content of your work.

In the following, I will chronologically mention all points that require amendment in my opinion but not mentioning every grammatical error as I am not an editor:

Cover Page

- According to the official website, Indian Institute of Technology Madras is written without a hyphen before Madras

Hyphen removed

- The word Limerick is duplicated in the name of the University of Limerick

A comma is added as it means University of Limerick in Limerick city (a comma was missing)

Abstract

- You write “Any analysis of Twitter data sets requires a word embedding model to transform words into numbers that can be processed using machine learning (ML) algorithms”. This is not correct. Other vectorization methods like bag-of-words can also be used. Please re-write.

The sentence was rewritten to the following statement:

Word embedding models are a key component in the analysis of Twitter data sets. As they are considered one of the important methods of transforming words into numbers that can be processed using machine learning (ML) algorithms

- “[…] be used in Twitter natural language processing (NLP) applications.” Do you mean “Twitter-based”?

The sentence was rewritten, and the word based was added

- “[…] superior performance in terms of number of recognized words and accuracy”. It is not clear to what accuracy refers. Make clear what you refer to, i.e., classification tasks on known data sets. Also, you did not measure accuracy but F1. Consider rephrasing.

The sentence was rewritten

the model is compared with existing models from the literature, and results demonstrate superior performance in terms of the number of recognized words and F1 score for classification tasks with known data sets, as well as the ability to work with emojis.

- “In all, we produce nine different versions of Arab2Vec […]”. Please write in past tense as this already done. This also applies to the rest of the manuscript.

Done

- “[…] and Experiments to validate […]”. Experiments should be written be written in lowercase.

fixed

Introduction

- Line 2: You reuse “huge” shortly after it has been used in the abstract. Please change for stylistic reasons.

Done

- Line 5: In American English, the first word after a colon is capitalized if it begins a complete sentence https://www.grammarly.com/blog/capitalization-after-colons/.

Fixed

- Line 8: Can you give an example of sentences that are different due to dialectics?

There are many words in Arabic that have different meanings depending on dialectics; for example, الجَدّ in Arabic means grandfather, while الجِدّ means diligence.

- Line 9: As mentioned above, you can use other techniques to vectorize text data besides word embeddings.

The sentence was rewritten

- Line 12: Help ML models with what?

The sentence is changed to needed by ML models.

- Line 26: You write about recognizing English words. How did you evaluate their recognition, which advantage brings it to recognize English words? Please elaborate why this is a feature and not an artefact.

Sometimes the tweets contain English symbols because the Arabic tweet writers may use English words or symbols inside the tweets, such as A+.

- Line 27: You tested your data on two data sets. However, my research showed that there are more data sets on which you can evaluate. Can you elaborate why you did not use them? Find some in section 5.2 from https://aclanthology.org/C16-1228.pdf, in section 4.1 of https://arxiv.org/pdf/2003.00104.pdf

We agree with the reviewer that there are many data sets and problems to test our developed model against. However, in this paper, we are just introducing the paper to the public and testing it with two different data sets just to prove the performance of the model. The authors provide the models for others to test in other different applications to evaluate them.

- Lines 40-46: Why do you write section in uppercase and why do your write “forth” in lowercase

All section words are changed to lowercase

- in line 42? Why are you missing omitting using an article for Section in this paragraph?

The paragraph has been revised

Literature Survey

This section is missing structure. I understood that you cited related research on Arabic word embeddings, highlighted the performance of these embeddings in several NLP-related tasks, and mentioned pre-processing strategies. However, motivation and structure of this section are unclear. For example, in lines 78-89, you wrote about NLP applications using embeddings and what kind of machine learning models were used to for these tasks, and how they performed. It is not clear why you are mentioning this, as I am expecting a paper on embeddings and not on sentiment analysis. Only later in the paper, when I read the section on evaluation, I understand that you were giving an overview on model performance with existing embeddings and evaluation strategy for this paper. Please make this clearer.

I suggest starting with a rough overview of the section by outlining the intention of the literature survey, namely listing similar work on word embeddings, existing evaluation and pre-processing strategies, and the current performance of these models as the reference for the author’s proposed embeddings.

A starting section for the literature survey is added to it

Finally, I am missing some mentioning of relevant other work. For example, https://aclanthology.org/L18-1577.pdf built a word embedding using even more tweets. I think it would be interesting to compare the performance to this embedding.

The proposed work by reviewer is not word embedding model! That the proposed model can be compared with

Also, I am missing the comparison to transformer-based approaches to embed words. https://arxiv.org/pdf/2003.00104.pdf and others have open-sourced their work on this. It would be particularly interesting to find out how your word2vec approach compares to the currently more popular transformer-based approach. This would help answer where word2vec approaches may be more senseful than transformer-based approaches. For your interest, I found this post, where they extract embeddings from pre-trained transformers: https://mccormickml.com/2019/05/14/BERT-word-embeddings-tutorial/#1-loading-pre-trained-bert

In this work, we propose a new model for word embeddings of Arabic twitter words. That is the focus of this research is to propose nine different models with different sizes to work in this research area. In the context of this research, a comparison is made to show the superior performance of the proposed nine models with respect to similar models in the literature. We want to thank the reviewer for his valuable comment, but we feel that comparing our work with pre-trained transformers could be a research extension problem for this work that we can study in future as it will require running more experiments and analysis to reach satisfied results.

Data pre-processing

- Line 141: How did you deduplicated? Did you check if strings were exactly the same or did you use some kind of fuzzy matching?

We mean here that the same tweets were written to a database we use for training the model. Thus, all we need is a string matching to remove such tweets

- Line 150-154: Why did you not use special tokens like <LINK> instead of the Arabic word for link which may be used in vastly different contexts? The same question applies to the Twitter mentions for, e.g., persons or brands.

In these steps, we are following the same steps of work done in literature (see for example, the AraVec paper); you will find they are using this approach. Also, the word <رابطويب> is rarely used in tweets. Also, the word <حسابشخصي>. Actually, both words are a combination of two Arabic words, which makes them a special token, as the reviewer suggests

- Did you not perform any other forms of pre-processing like normalization?

All steps of pre-processing were described in the paper. We don’t perform normalization to words as we wanted the model to be trained on different words as they are and not trained only on the root of words.

- How did you tokenize your data?

With the tree bank word tokenizer library in Python.

Effort in these steps

- Line 181: Do you mean the window function? Does this mean, you examine 300 words before and after the word of interest? To me, word count threshold sounds like a trimming rule to remove words that occur less than 300 times. Why is this threshold different for both models? Please be more accurate. How large are the embeddings, how large was the context window? Ideally, list all hyperparameters and put them into the appendix.

The Word2Vec model includes a parameter called the "word count threshold," which helps to eliminate infrequent words from the training data. This is important to reduce noise and improve the quality of the resulting word vectors. The threshold can be set to a minimum number of occurrences, below which words are discarded from the training data. In our analysis, we used different word count thresholds of 100 and 300 to generate models of different sizes that may be suitable for different applications or users. We also found that even rare or infrequent words that occurred at least

- Line 184: You treat the whole corpus as tri-grams? Isn’t it more practical to only consider tri-grams when words frequently appear in bi- or tri-grams like here: https://radimrehurek.com/gensim/models/phrases.html#module-gensim.models.phrases

In addition, we aimed to reduce human involvement and allow the model more freedom by not limiting it based on smaller n-grams. This was important because Arabic words have multiple trigrams that do not correspond to bigrams, such as the names of the prophet Muhammad's companions, as well as other Arabic names and hashtags. Limiting the model to smaller n-grams could potentially affect our results. However, we also created a smaller model based only on unigrams to cater to different users and applications

Model evaluation and comparison

- Line 194:

---

## [Decision Letter · Decision Letter 1]

6 Sep 2023

PONE-D-23-01033R1Arab2Vec: An Arabic word embedding model for use in Twitter NLP applicationsPLOS ONE

Dear Dr. Ryan,

Thank you for submitting your manuscript to PLOS ONE. After careful consideration, we feel that it has merit but does not fully meet PLOS ONE’s publication criteria as it currently stands. Therefore, we invite you to submit a revised version of the manuscript that addresses the points raised during the review process.

We look forward to receiving your revised manuscript.

Kind regards,

Junaid Rashid, Ph.D

Academic Editor

PLOS ONE

Reviewers' comments:

Reviewer's Responses to Questions

**Comments to the Author**

1. If the authors have adequately addressed your comments raised in a previous round of review and you feel that this manuscript is now acceptable for publication, you may indicate that here to bypass the “Comments to the Author” section, enter your conflict of interest statement in the “Confidential to Editor” section, and submit your "Accept" recommendation.

Reviewer #1: (No Response)

Reviewer #3: (No Response)

2. Is the manuscript technically sound, and do the data support the conclusions?

Reviewer #1: Yes

Reviewer #3: No

3. Has the statistical analysis been performed appropriately and rigorously? 

Reviewer #1: Yes

Reviewer #3: No

4. Have the authors made all data underlying the findings in their manuscript fully available?

Reviewer #1: Yes

Reviewer #3: No

5. Is the manuscript presented in an intelligible fashion and written in standard English?

Reviewer #1: Yes

Reviewer #3: No

6. Review Comments to the Author

Reviewer #1: All comments, with except of number 6, were answered.

The authors need to provide specifications of the system used to train the models (GPU, processor, RAM, etc…), or at least give an idea of the computing power they had -> is it a powerful cluster or a typical laptop?

Reviewer #3: 1. The provided GitHub link does not grant access to the models and code.

2. The paper writing requires improvement. For instance, there are sentences like, "• is an open-source model trained on 186M Tweets;" and "can recognize ..., " Please correct all bullet point sentences.

3. The model is not open-source, and the link does not function.

4. Under "Literature Review", discuss the challenges of previous methods and explain how your approach addresses these issues. Including a comparison table detailing previous methods would enhance this section.

5. In the "Dataset" section, clarify how the datasets were collected and which keywords were utilized.

6. Is the dataset available online for experimentation?

7. Please explain the rationale behind not using publicly available datasets for your experiments. Additionally, compare your results with those from previous methods, especially on public datasets.

8. The paper English needs improvement. Phrases such as "the total number…" make it challenging to understand. The paper contains multiple errors, making it difficult to read.

9. The paper lacks a statistical analysis section.

10. The results are exclusively compared with models like CBOW, Skip-gram, and Skip-gram NS with Arab2Vec. It would be beneficial to also contrast the outcomes with other model types, such as deep learning and transformer-based models.

11. The paper novelty appears restricted. Consider creating a framework diagram to explain the comprehensive steps of your model.

Based on the concerns highlighted above, the paper does not meet the criteria for publication, leading to my decision to reject it.

7. PLOS authors have the option to publish the peer review history of their article (what does this mean?). If published, this will include your full peer review and any attached files.

Reviewer #1: No

Reviewer #3: No

---

## [Author Response · Author response to Decision Letter 2]

18 Oct 2023

All comments addressed. We had some extra items for the editor in the cover letter, which are also here:

We have addressed all reviewer comments as noted below. There are two issues that I would like to draw to your attention, both of which are related to Reviewer #3. The first is something I have already spoken to you about, but Reviewer #3 doesn’t have the context as they are new. Specifically, we can’t make the raw data available for copyright reasons – they are tweets scrapped from Twitter. The second issue concerns their comment about ungrammatical passages, specifically those in bulleted lists. We have added an explanation for why the text is grammatically correct, along with a quote from and a link to the relevant APA-style webpage.

If there is some stylistic reason that PLoS One would prefer us not to use that style, then, of course, we are happy to oblige.

---

## [Decision Letter · Decision Letter 2]

2 Jan 2024

PONE-D-23-01033R2Arab2Vec: An Arabic word embedding model for use in Twitter NLP applicationsPLOS ONE

Dear Dr. Ryan,

Thank you for submitting your manuscript to PLOS ONE. After careful consideration, we feel that it has merit but does not fully meet PLOS ONE’s publication criteria as it currently stands. Therefore, we invite you to submit a revised version of the manuscript that addresses the points raised during the review process.

We look forward to receiving your revised manuscript.

Kind regards,

Junaid Rashid, Ph.D

Academic Editor

PLOS ONE

Reviewers' comments:

Reviewer's Responses to Questions

**Comments to the Author**

1. If the authors have adequately addressed your comments raised in a previous round of review and you feel that this manuscript is now acceptable for publication, you may indicate that here to bypass the “Comments to the Author” section, enter your conflict of interest statement in the “Confidential to Editor” section, and submit your "Accept" recommendation.

Reviewer #2: (No Response)

Reviewer #3: (No Response)

2. Is the manuscript technically sound, and do the data support the conclusions?

Reviewer #2: Yes

Reviewer #3: No

3. Has the statistical analysis been performed appropriately and rigorously? 

Reviewer #2: Yes

Reviewer #3: No

4. Have the authors made all data underlying the findings in their manuscript fully available?

Reviewer #2: No

Reviewer #3: (No Response)

5. Is the manuscript presented in an intelligible fashion and written in standard English?

Reviewer #2: Yes

Reviewer #3: (No Response)

6. Review Comments to the Author

Reviewer #2: Please try your best to thoroughly go through the reviewers points and be as consistent as possible in your manuscript.

Reviewer #3: I have identified several issues with the paper.

1. The authors did not adequately highlight the changes made and did not provide a response letter addressing each comment. However, upon reviewing the paper, I found that most of the problems in the manuscript remain unresolved such as comments #5, 7, 10, and 11.

2. The paper now includes Table 1, representing the number of recognized words, recognizing emojis, and NS variants. However, it lacks a discussion of the challenges posed by previous methods and how the proposed approach addresses these issues.

3. Regarding previous Comment #5, the "Dataset" section requires clarification on how the datasets were collected and which keywords were utilized. The information about the keywords used is still missing.

4. Comment # 7- Please explain the rationale behind not using publicly available datasets for your experiments. Additionally, compare your results with those from previous methods, especially on public datasets for fair comparisons.

5. Comment # 10- The results are exclusively compared with models like CBOW, Skip-gram, and Skip-gram NS with Arab2Vec. It would be beneficial to also contrast the outcomes with other model types, such as deep learning and transformer-based models.

6. Comment # 11, Consider creating a framework diagram to explain the comprehensive steps of your model.

As per aforementioned comments, the manuscript is still not suitable for acceptance. It does not adhere to the PLOS ONE criteria, specifically regarding the performance of experiments, statistics, and other analyses to a high technical standard, along with a lack of sufficient detail in their description.

7. PLOS authors have the option to publish the peer review history of their article (what does this mean?). If published, this will include your full peer review and any attached files.

Reviewer #2: **Yes: **Auss Abbood

Reviewer #3: No

---

## [Author Response · Author response to Decision Letter 3]

2 May 2024

Reviewer 2 response notes

Introduction

- Lines 5 and 6: Grammatical errors.

Grammar of the sentence is revised. The new sentence is highlighted for reviewers.

Added to manuscript: However, working with the Arabic language is challenging for two reasons: first, there is a low number of data sets and pre-trained models, and second, the language itself is more complex to work with than Western languages.

- Line 6: I asked in the first round of revision to give examples of dialectics in Arabic. Please explain this also in the manuscript not only in your response letter. Also, dialectic may be the wrong word. Do you mean dialects? Dialectic is a way of philosophical reasoning by assuming contrary positions.

The word diacritic. For example Word: سَكَرَ (pronounced as "sakara") Without diacritics: سكر

Meaning with diacritics: He/She became drunk (past tense)

Meaning without diacritics: Sugar (noun)

The word is fixed, and an example is added to the manuscript.

Definition: https://www.merriam-webster.com/dictionary/diacritic

- Line 11: Help ML models do what? It is clear to me what you mean but please make it clear for the broad audience.

Fixed

Added to manuscript: These vectors of numbers capture the semantic properties of words, which can help ML models do better job in their required tasks.

Literature Survey

- Line 56, 63, and 65: Why do you not use something like \citet to use the author’s name. Reading, for example, “was introduced by [11]” feels awkward.

The journal style is a numbering style. However, referencing is revised to make sentences more clear.

- Line 90: You do not thoroughly introduce abbreviations before using them. Please always write a word out before using abbreviations. Latex, which you seem to be using, offers a range of solutions to automate this.

All abbreviations have been revised.

- You write some titles with capital letters like “Literature Survey” and some without like “Model evaluation and comparison”. Please use one style.

The style with capital letters is used. All titles are revised according to the reviewer comment

Background Information

- Line 105: Please provide evidence that syntactic information is also encoded in vector representations.

The sentence was rephrased, and syntactic information was removed. Syntactic information is not discussed in this work and is out of scope.

- Is Figure 1 really made by you? If not, you need to quote your source and check if you have the rights to use this figure.

Figure 1 is introduced in the original paper about CBOW and Skip-gram. We have added a reference number in the figure label to the first paper that introduced the figure. The figure is widely used in different papers in literature in the same way we use it, i.e., citing the original paper of CBOW and Skip-gram.

Data pre-processing

- How did you recognize whether an Tweet was written in Arabic? Please include your answer into the manuscript.

The tweets data retrieved from twitter have a column field called language. We compare this field with the Arabic language. After this, a step is applied to remove duplicate tweets or URLs, etc.

A sentence to clarify this is added to text.

Model evaluation and comparison

- Line 192: Redundant explanation of what the total number of words recognized is.

Tightened up the language.

- Line 202, 203, 206, 212, 221: Grammatical errors.

Grammar was revised, and some errors were corrected

- GitHub link is not visible and should not be placed in the footnote like this.

Github is removed from the footnote and embedded into the text to make it clearer for the reader.

- Line 223: Why do you choose to evaluate a word not recognized by ArabWordVec? You answered that you want to show the poorer performance of recognizing words of the other models. However, you already made clear that Arab2Vec recognizes more words. This suffices. This section is about assessing word similarity. Please pick another word.

Respectfully, we disagree. Including a word that two models recognise gives insight into the two more sophisticated models. If we only used words that ArabWordVec can recognise it will be a limited comparison.

- Line 224: Why do you write Figure (4) and everywhere else you do not write the Figure number in parentheses?

Just a typo, and fixed all figures’ numbering are consistent

- Line 226: Why is there so much white space?

White space removed.

Figure 4: The emojis are close to not being readable. As per my first review, I encourage you to ditch the black background and choose a white background. Also, your figures look cramped, as if someone resized them without keeping the original aspect of the image. The letters look way too wide. So do the emojis. They should be round but appear oval. Please change this.

We believe the emojis look clear, particularly when zooming on the image. This final colouring results from suggestions from earlier rounds of reviews.

Experimental comparison of the models

- Line 274, 276, 277: Grammatical error. Also, you write Table with capital letters. Before you wrote it with small letters. Please stay consistent. I prefer it the capitalized version.

All tables are referenced with capital letters

- Table 3. The English word “bus” is linked to the Arabic plural of bus. Please change the English word to busses or the Arabic word from albaaSaat to albaaS.

Agreed. The word fixed. Please check Table 4 as we added a table.

- Table 3: The word Arabic word for Muhammad looks like the Arabic letters “kmd”. Is it due to the font that the first letter looks like a “k” instead of a “muHa”?

In Arabic, there are different styles of writing. One can write محمد or but the م above ح like in the manuscript. Native Arabic speakers will understand. However, it is generated like this from the library we are using.

- Line 285: Same as above, you inconsistently abbreviate or write out words.

Abbreviations are added

- Line 286: Grammatical error.

All Grammatical errors are revised

- Figure 8. The word in the social media cluster that receives and extra explanation is impossible to read for me. It seems half-cut off.

We have added a clear table that mentions each word in every cluster, as, in the figure, words may interfere with each other and not be clear.

Line 287 and 288: Grammatical error.

All grammatical errors are revised

Conclusion

- Line 301 and 312: Grammatical error.

Revised

- Table 5. Missing period after text.

ِِAdded

- Missing explanation what NS, LR, SVC, and GB mean. See comments on abbreviations from above.

Abbreviations are added into the text the first time these algorithms are mentioned.

- You inconsistently use 1, 2, or 3 digits after the period.

2 digits are used except if the second digit after the period is zero.

Why do you write all performances of Arab2Vec Skip Gram in italics. With SVC, e.g., it is does not have the best performance.

Fixed only best accuracies are italic in the table

- Table 6: See Table 5. The table needs to be understood without looking into the manuscript. This is not possible. What is ML what is DL? Be specific.

ML stands for machine learning models while DL stands for deep learning models. This is explained in the Table caption.

- Table 8 and 9 miss indications of the best performing Word2Vec models

Added to tables (italic indication of the best-performing Word2Vec models)

- and in Table 8 you are missing a period in the table text.

Period is added.

Missed points from last review

Some points may repeat compared to points made above.

- Example for dialects still not in manuscript.

Example is added into manuscript.

- Inconsistencies in lower- and upper cases in section titles persist.

Fixed and revised in all manuscript.

- The authors answered how they tokenized their data but did not describe it in the manuscript.

Tree bank tokenizer library is used in Python (Added to manuscript)

All technical details should be listed in the manuscript. Readers should not require to open a GitHub repository to understand what you did. The same applies to hyper parameters of the word2vec models and discussions on windows size and reasons for selecting tri-grams.

The reason for using tri-grams was explained in the previous round of reviews

In addition, we aimed to reduce human involvement and allow the model more freedom by not limiting it based on smaller n-grams. This was important because Arabic words have multiple trigrams that do not correspond to bigrams, such as the names of the prophet Muhammad's companions, as well as other Arabic names and hashtags. Limiting the model to smaller n-grams could potentially affect our results. (added to manuscript)

- You still use vertical lines in your tables which I discouraged you from using giving a reference.

We tried this, but is made the tables look messy.

- Even gray background is hard to read. In the reference I gave, it is clear that much brighter background is required for readability like white. Please follow the instruction in the reference or give a counter reference but do not ignore this point.

Respectfully, we disagree. There is sufficient contrast.

- Please include information on using k-means clustering in the manuscript.

k-means is an unsupervised machine learning algorithm. In this work we don’t use it.

- There is a misunderstanding on which highlighting method should be used in Tables to indicate the best performing model. Using bold is perfectly fine for me it should just be consistent. In this second version, Table 8 and 9 are missing such highlighting.

Fixed in this version, italic is used for the highest accuracy to show the best performance

- Digits are still not rounded to two digits.

Fixed some numbers were reported to 3 digits. Now they are rounded to two digits. In some numbers, the second digit is zero, like .90. These are reported as .9, which makes more sense.

- There is a mess on whether you show result on the validation or test set. You still write validation set in Table 5, 8, and 9. Please fix this.

We agree with the reviewer there was an error that is now fixed. All results are shown on test data that are not used during training the model.

- Please describe your classification set up. How did you split your data. Was your dataset balanced? As mentioned above, it is not enough to describe these details only in code. Everything should be at one place.

I suppose the reviewer means the ASTD and COVID datasets. For splitting the data, a shuffle and split data function is used.

Reviewer #3: I have identified several issues with the paper.

1. The authors did not adequately highlight the changes made and did not provide a response letter addressing each comment. However, upon reviewing the paper

We thank the reviewer for his valuable comment in this round of revision. A response letter is provided along with a version of the manuscript highlighted with changes.

I found that most of the problems in the manuscript remain unresolved such as comments #5, 7, 10, and 11.

2. The paper now includes Table 1, representing the number of recognized words, recognizing emojis, and NS variants. However, it lacks a discussion of the challenges posed by previous methods and how the proposed approach addresses these issues.

There is a dedicated section in the introduction to discuss the challenges of previous Arabic word embedding models and discuss the novel contributions of the proposed work here. This section is usually placed in the last paragraph of the introduction.

Table 1 and other Tables are to prove our contributions that are already explained in introduction

The following sentence is added to the manuscript to answer the reviewer comment.

In summary, prior Arabic word embedding models had multiple challenges, including a limited set of recognized words, an inability to identify emojis, English words embedded in Arabic tweets, and an inability to identify phrases associated with the COVID-19 pandemic. Our models resolve these issues.

3. Regarding previous Comment #5, the "Dataset" section requires clarification on how the datasets were collected and which keywords were utilized. The information about the keywords used is still missing.

Information is added on how the Dataset was collected. In this work, we are collecting all Arabic tweets, so we are not using keywords. Also, using keywords for collecting data from twitter proved to produce wrong datasets as it may collect a tweet with a similar word but not in the topic.

In this work, we are using the language column available in the retrieved tweets that identifies the language of the tweet.

4. Comment # 7- Please explain the rationale behind not using publicly available datasets for your experiments. Additionally, compare your results with those from previous methods, especially on public datasets for fair comparisons.

We clarified that we are using publicly available data sets and gave links to them.

5. Comment # 10- The results are exclusively compared with models like CBOW, Skip-gram, and Skip-gram NS with Arab2Vec. It would be beneficial to also contrast the outcomes with other model types, such as deep learning and transformer-based models.

The results are compared with two similar models. Our proposed model, Arab2Vec, is compared with AraWordVec and AraVec, which are two similar models introduced in the literature similar to our model. All three models are deep learning. Also, as explained in the previous rebuttal letter, transformer-based models are out of the scope of this work. Future work may include a comparison between word embedding models and transformer-based models. However, transformer-based models will be more complex to use for this task and will require high computational costs. Here, we provide nine different models to give the reader the ability to choose the size suitable from a computational point of view. This will not be the case in transformer-based models.

6. Comment # 11, Consider creating a framework diagram to explain the comprehensive steps of your model.

A framework is added to paper

---

## [Decision Letter · Decision Letter 3]

13 Sep 2024

PONE-D-23-01033R3Arab2Vec: An Arabic word embedding model for use in Twitter NLP applicationsPLOS ONE

Dear Dr. Ryan,

Thank you for submitting your manuscript to PLOS ONE. After careful consideration, we feel that it has merit but does not fully meet PLOS ONE’s publication criteria as it currently stands. Therefore, we invite you to submit a revised version of the manuscript that addresses the points raised during the review process.

**ACADEMIC EDITOR: **The authors have incorporated the previous comments of the reviewers. However, there are still some critical points from Reviewer 3 that need to be addressed.

We look forward to receiving your revised manuscript.

Kind regards,

Junaid Rashid, Ph.D

Academic Editor

PLOS ONE

Reviewers' comments:

Reviewer's Responses to Questions

**Comments to the Author**

1. If the authors have adequately addressed your comments raised in a previous round of review and you feel that this manuscript is now acceptable for publication, you may indicate that here to bypass the “Comments to the Author” section, enter your conflict of interest statement in the “Confidential to Editor” section, and submit your "Accept" recommendation.

Reviewer #3: (No Response)

2. Is the manuscript technically sound, and do the data support the conclusions?

Reviewer #3: (No Response)

3. Has the statistical analysis been performed appropriately and rigorously? 

Reviewer #3: (No Response)

4. Have the authors made all data underlying the findings in their manuscript fully available?

Reviewer #3: (No Response)

5. Is the manuscript presented in an intelligible fashion and written in standard English?

Reviewer #3: (No Response)

6. Review Comments to the Author

Reviewer #3: Authors incorporated my previous comments but still some points need to be addressed.

1. Authors mentioned that language column, It’s not clear which specific Arabic tweets were selected.

2. "Our proposed model has several advantages. Specifically, it"-> Write 3-4 main contributions of the paper.

4. Add a Discussion section.

5. The paper English needs improvement.

7. PLOS authors have the option to publish the peer review history of their article (what does this mean?). If published, this will include your full peer review and any attached files.

Reviewer #3: No

---

## [Author Response · Author response to Decision Letter 4]

8 Oct 2024

Dear Editor-in-chief (PLOS ONE):

I am pleased to submit our revised article titled “Arab2Vec: An Arabic word embedding model for use in Twitter NLP applications’’ by Abdelrahman Hamdy, Ayman Youssef and Conor Ryan for your consideration.

There were only comments from Reviewer #3 on this version. These were addressed as below.

We want to thank the reviewer for his valuable comments in enhancing the manuscript.

Reviewer #3: Authors incorporated my previous comments but still some points need to be addressed.

1. Authors mentioned that language column, It’s not clear which specific Arabic tweets were selected.

There was no specific way for choosing the Arabic tweets we use in different experiments like clustering of named entity. We reported the words we selected which are similar to previous papers published in literature. However, all the models were tested using the same selected tweets for the fairness of comparison. A number of words are chosen to represent each cluster.

A similar sentence is added to manuscript for clearness.

2. "Our proposed model has several advantages. Specifically, it"-> Write 3-4 main contributions of the paper.

The contributions of the work are summarized now in four main points according to the reviewer comment.

4. Add a Discussion section.

A discussion section is added to manuscript.

5. The paper English needs improvement.

Some language modifications are made to the manuscripts to improve English. The paper was revised by one of the authors who is a native English speaker. Also, software language checks were made.

Grammarly gives the current draft a score of 99/100

---

## [Decision Letter · Decision Letter 4]

10 Jan 2025

PONE-D-23-01033R4Arab2Vec: An Arabic word embedding model for use in Twitter NLP applicationsPLOS ONE

Dear Dr. Ryan,

Thank you for submitting your manuscript to PLOS ONE. After careful consideration, we feel that it has merit but does not fully meet PLOS ONE’s publication criteria as it currently stands. Therefore, we invite you to submit a revised version of the manuscript that addresses the points raised during the review process.

**ACADEMIC EDITOR: **The authors have addressed the previous comments from the reviewers. The paper is accepted for publication with minor revisions, and the authors must address the remaining minor comments from the reviewer.

We look forward to receiving your revised manuscript.

Kind regards,

Junaid Rashid, Ph.D

Academic Editor

PLOS ONE

Journal Requirements:

Reviewers' comments:

Reviewer's Responses to Questions

**Comments to the Author**

1. If the authors have adequately addressed your comments raised in a previous round of review and you feel that this manuscript is now acceptable for publication, you may indicate that here to bypass the “Comments to the Author” section, enter your conflict of interest statement in the “Confidential to Editor” section, and submit your "Accept" recommendation.

Reviewer #4: All comments have been addressed

2. Is the manuscript technically sound, and do the data support the conclusions?

Reviewer #4: Yes

3. Has the statistical analysis been performed appropriately and rigorously? 

Reviewer #4: Yes

4. Have the authors made all data underlying the findings in their manuscript fully available?

Reviewer #4: Yes

5. Is the manuscript presented in an intelligible fashion and written in standard English?

Reviewer #4: No

6. Review Comments to the Author

Reviewer #4: Thank you for addressing the comments. A few minor changes can further improve your manuscript as follows:

- Write the contributions in a standard manner. For instance, avoid starting with phrases like is an open-source model that was.... Instead, rephrase as Our model is an open-source.... Revise all contributions accordingly.

- Refer to the section names in the last paragraph of the introduction for easy navigation. For example: The paper is organized as follows: Section 1 presents a literature survey, Section 2 describes the data collection process, and Section 3 discusses the data preprocessing conducted on the collected dataset. Ensure to use clickable references for section names.

- Improve the quality of the diagrams. Use high DPI versions, as the current ones appear blurry, particularly the text.

- Consider adding recent models in the discussion or future work sections. For example, mention that Large Language Models can be fine-tuned to produce embeddings for these tasks.

- Overall, improve the English language throughout the manuscript. For instance, in the Abstract, revise sentences like The authors of this article provide Arab2Vec as an open-source project for users to employ in any research-related work. to We provide Arab2Vec as an open-source project for researchers to use in various applications.

7. PLOS authors have the option to publish the peer review history of their article (what does this mean?). If published, this will include your full peer review and any attached files.

Reviewer #4: No

---

## [Author Response · Author response to Decision Letter 5]

7 Feb 2025

Dear Editor-in-chief (PLOS ONE):

I am pleased to submit our revised article titled “Arab2Vec: An Arabic word embedding model for use in Twitter NLP applications’’ by Abdelrahman Hamdy, Ayman Youssef and Conor Ryan. We thank the reviewer for their comments and believe that we have addressed all of them as below.

Write the contributions in a standard manner. For instance, avoid starting with phrases like is an open-source model that was.... Instead, rephrase as Our model is an open-source.... Revise all contributions accordingly.

The section describing the work contributions is rewritten according to the reviewer comments. All contributions were rephrased.

“The proposed model has several advantages over other models in the literature. These can be summarized as follows: 32• It is an open-source model trained on 186M tweets and, therefore, can recognize

more words (2,027,042, approximately 33%) than previous models;

• It recognizes COVID-19-related words and emojis with higher accuracy as it was

trained on a newer data set; 36• It has a variant trained using negative sampling. This is an advantage for our work, as we will see from the results that this variant achieves higher accuracy than the previous model.

• It exhibits better performance than the two state-of-the-art models from the

literature (AraVec [9], AraWordVec [10] ) on the two tested data sets (COVID-19,

ASTD).”

- Refer to the section names in the last paragraph of the introduction for easy navigation. For example: The paper is organized as follows: Section 1 presents a literature survey, Section 2 describes the data collection process, and Section 3 discusses the data preprocessing conducted on the collected dataset. Ensure to use clickable references for section names.

The last paragraph of the introduction was modified depending on the reviewer comments

Section 1, which contains a literature survey, before moving to Section 2, which describes the data collection process, and Section 3, which discusses the pre-processing conducted on that set. Section 4 presents a qualitative comparison between the model and two other well-known models from the literature using publicly available data. Next, Section 5 presents a quantitative experimental comparison between the proposed model and models from the literature. Finally, Section 6 provides some conclusions and describes some future work.

- Improve the quality of the diagrams. Use high DPI versions, as the current ones appear blurry, particularly the text.

All images of the manuscript were revised. We are not able to get clearer images from software, but some images just needed resizing to make them clearer, while other images were enhanced using various tools from the internet. However, clear description of figures with tables for each word has been added into manuscript for clarity in previous review rounds.

- Consider adding recent models in the discussion or future work sections. For example, mention that Large Language Models can be fine-tuned to produce embeddings for these tasks.

A new paragraph has been added to the discussion section concerning the capabilities of large language models.

“An alternate approach would be to employ a Large Language Model (LLM) to perform this task. In particular, the ability of LLMs to create context-aware representations is attractive. However, a dedicated word embedding model like the one proposed here can reasonably be expected to be more efficient and resource-friendly than fine-tuning a large language model for word embedding tasks. Future work will explore this trade-off further.”

- Overall, improve the English language throughout the manuscript. For instance, in the Abstract, revise sentences like The authors of this article provide Arab2Vec as an open-source project for users to employ in any research-related work. to We provide Arab2Vec as an open-source project for researchers to use in various applications.

The entire manuscript was revised to add clarity to the text. Several sentences were rewritten, other were modified for clarity.

---

## [Decision Letter · Decision Letter 5]

11 Mar 2025

PONE-D-23-01033R5Arab2Vec: An Arabic word embedding model for use in Twitter NLP applicationsPLOS ONE

Dear Dr. Ryan,

Thank you for submitting your manuscript to PLOS ONE. After careful consideration, we feel that it has merit but does not fully meet PLOS ONE’s publication criteria as it currently stands. Therefore, we invite you to submit a revised version of the manuscript that addresses the points raised during the review process.

**ACADEMIC EDITOR: **A few minor issues remain that should be addressed to ensure the clarity and overall quality of the final version. In particular, the quality of Figure 2 needs improvement, as the current version lacks the visual clarity expected for publication. The figure should be recreated to enhance its readability, method overall steps and presentation. Furthermore, although the introduction has been improved, the section discuss the paper’s contributions would benefit from further refinement. More clearly discussing the key findings, remove the references inside contributions and emphasizing the novelty of the work will help readers better understand the significance and impact of the research. Finally, the manuscript should be carefully proofread to correct any remaining language issues and ensure consistency in style and formatting throughout.

We look forward to receiving your revised manuscript.

Kind regards,

Junaid Rashid, Ph.D

Academic Editor

PLOS ONE

Journal Requirements:

Reviewers' comments:

Reviewer's Responses to Questions

**Comments to the Author**

1. If the authors have adequately addressed your comments raised in a previous round of review and you feel that this manuscript is now acceptable for publication, you may indicate that here to bypass the “Comments to the Author” section, enter your conflict of interest statement in the “Confidential to Editor” section, and submit your "Accept" recommendation.

Reviewer #4: All comments have been addressed

2. Is the manuscript technically sound, and do the data support the conclusions?

Reviewer #4: Yes

3. Has the statistical analysis been performed appropriately and rigorously? 

Reviewer #4: Yes

4. Have the authors made all data underlying the findings in their manuscript fully available?

Reviewer #4: Yes

5. Is the manuscript presented in an intelligible fashion and written in standard English?

Reviewer #4: Yes

6. Review Comments to the Author

Reviewer #4: Thank you for addressing the comments. Please address these following minor comments:

- Fig. 2 is not acceptable in the current form due to its low quality. You could use some online drawing tools such as draw.io to make it more visible/clear.

- Contributions in the Introduction is a core of the manuscript. Revise the writing of introduction, specifically, contributions to convey your ideas/findings/results with more clarity.

7. PLOS authors have the option to publish the peer review history of their article (what does this mean?). If published, this will include your full peer review and any attached files.

Reviewer #4: No

---

## [Author Response · Author response to Decision Letter 6]

10 Jun 2025

Dear Editor-in-chief (PLOS ONE):

I am pleased to submit our revised article titled “Arab2Vec: An Arabic word embedding model for use in Twitter NLP applications’’ by Abdelrahman Hamdy, Ayman Youssef and Conor Ryan for your consideration. We have addressed the final issues noted by the reviewer as follows:

In particular, the quality of Figure 2 needs improvement, as the current version lacks the visual clarity expected for publication. The figure should be recreated to enhance its readability, method overall steps and presentation.

Figure 2 was recreated to give more clarity to the steps involved in the work.

Furthermore, although the introduction has been improved, the section discusses the paper’s contributions would benefit from further refinement. More clearly discussing the key findings, remove the references inside contributions and emphasizing the novelty of the work will help readers better understand the significance and impact of the research.

This part has been completely rewritten, although we have kept the citations in place as this is where these data sets are mentioned.

Finally, the manuscript should be carefully proofread to correct any remaining language issues and ensure consistency in style and formatting throughout.

The manuscript has been proofread once more and checked with Grammarly, which returns a score of 99/100.

---

## [Decision Letter · Decision Letter 6]

1 Jul 2025

Arab2Vec: An Arabic word embedding model for use in Twitter NLP applications

PONE-D-23-01033R6

Dear Dr. Ryan,

We’re pleased to inform you that your manuscript has been judged scientifically suitable for publication and will be formally accepted for publication once it meets all outstanding technical requirements.

Kind regards,

Junaid Rashid, Ph.D

Academic Editor

PLOS ONE

Reviewers' comments:

Reviewer's Responses to Questions

**Comments to the Author**

1. If the authors have adequately addressed your comments raised in a previous round of review and you feel that this manuscript is now acceptable for publication, you may indicate that here to bypass the “Comments to the Author” section, enter your conflict of interest statement in the “Confidential to Editor” section, and submit your "Accept" recommendation.

Reviewer #4: All comments have been addressed

2. Is the manuscript technically sound, and do the data support the conclusions?

Reviewer #4: Yes

3. Has the statistical analysis been performed appropriately and rigorously? 

Reviewer #4: Yes

4. Have the authors made all data underlying the findings in their manuscript fully available?

Reviewer #4: Yes

5. Is the manuscript presented in an intelligible fashion and written in standard English?

Reviewer #4: Yes

6. Review Comments to the Author

Reviewer #4: The author has addressed all the reviewers' comments, and I recommend accepting this article in its current form.

7. PLOS authors have the option to publish the peer review history of their article (what does this mean?). If published, this will include your full peer review and any attached files.

Reviewer #4: No

---

## [Editor Report · Acceptance letter]

PONE-D-23-01033R6

PLOS ONE

Dear Dr. Ryan,

I'm pleased to inform you that your manuscript has been deemed suitable for publication in PLOS ONE. Congratulations! Your manuscript is now being handed over to our production team.

Kind regards,

on behalf of

Dr. Junaid Rashid

Academic Editor

PLOS ONE